# Cross-Country Student Perceptions about Online Medical Education during the COVID-19 Pandemic

**DOI:** 10.3390/ijerph19052840

**Published:** 2022-02-28

**Authors:** Tomoya Suzuki, Anju Murayama, Yasuhiro Kotera, Divya Bhandari, Yuki Senoo, Yuta Tani, Kayo Harada, Ayumu Kawamoto, Satomi Sato, Toyoaki Sawano, Yasushi Miyata, Masaharu Tsubokura, Tetsuya Tanimoto, Akihiko Ozaki

**Affiliations:** 1Medical Governance Research Institute, Takanawa, Minato-ku, Tokyo 1087505, Japan; rayordeal3@gmail.com (D.B.); senooyuki0821@gmail.com (Y.S.); tyuta0430@gmail.com (Y.T.); kayoharada.0615@gmail.com (K.H.); ayumu0210604@gmail.com (A.K.); 18a1051@g.iuhw.ac.jp (S.S.); tsubokura-tky@umin.ac.jp (M.T.); tetanimot@me.com (T.T.); ozakiakihiko@gmail.com (A.O.); 2School of Medicine, Akita University, 1-1-1 Hondo, Akita 0108543, Japan; 3School of Health Sciences, University of Nottingham, Nottingham NG7 2HA, UK; yasuhiro.kotera@nottingham.ac.uk; 4Department of Surgery, Jyoban Hospital of Tokiwa Foundation, Iwaki 9728322, Japan; toyoakisawano@gmail.com; 5Department of Primary Care and Community Health, Aichi Medical University, Nagakute 4801195, Japan; ymymiyata@gmail.com; 6Department of Radiation Health Management, Fukushima Medical University School of Medicine, Fukushima 9601247, Japan; 7Department of Internal Medicine, Navitas Clinic Kaswasaki, Kawasaki 2100007, Japan; 8Department of Breast Surgery, Jyoban Hospital of Tokiwa Foundation, Iwaki 9728322, Japan

**Keywords:** undergraduate, COVID-19, pandemic, medical education, SARS-CoV-2, distance-learning

## Abstract

(1) Introduction: Most educational institutions around the world have shifted from traditional face-to-face to online education amid COVID-19. This change may particularly impact medical students, whose education is heavily influenced by clinical learning experiences. Accordingly, we investigated medical students’ perceptions about positive and negative aspects of online medical education in Japan and overseas during the COVID-19 pandemic. (2) Methods: In-depth online interviews were conducted among 13 Japanese medical students and five medical students from Slovakia, Norway, and Hungary. Interviews were conducted from 23rd September to 3rd October 2020 using the snowball sampling method. Questions were focused on five main areas: Q1 the type of online education; Q2 advantages and disadvantages of online education; Q3 any changes in the relationship with teachers, friends, and family; Q4 any opinions about further improvements in online education; and Q5 any needs for affiliation with a particular university. Then thematic analysis was conducted. (3) Results: The results of the thematic analysis revealed the following four themes that represent the positive and negative aspects of online medical education; Theme 1: Timesaving and Flexibility; Theme 2: Technical problems and lack of digital skills; Theme 3: Unstandardized teaching skills; Theme 4: Lack of experience beyond medical school lectures. (4) Conclusions: While online education was found useful in terms of saving time and creating a flexible learning environment, many important drawbacks were noted such as internet and computer problems and unstandardized teaching skills, and lack of quality assurance. In addition, experiences outside the classroom such as making relationships with faculty and friends, conducting research and participating in extracurricular activities were missed, which they normally enjoy in college life.

## 1. Introduction

Clinical experience in medical education is a fundamental step for undergraduate medical students to learn practical patient care [1]. However, along with the advent of Coronavirus Disease 2019 (COVID-19), governments were forced to implement strict policy including the closure of public spaces, travel restrictions, national lockdowns, and school closure [2,3]. These restrictions were also extended to medical schools across the world, and medical students were subjected to many restrictions in their private and educational settings, such as being forced to change their normal learning environment, being isolated in their house or dormitory, etc. [4,5,6].

In Japan, since the first COVID-19 case was found on 16 January 2020 [7], the number of confirmed cases gradually increased, and cumulative confirmed cases reached more than 250 as of 2 March 2020 [8]. Due to pressure from the failure of the Diamond Princess [9], the Japanese government enforced a strict policy to close schools as early as 2 March 2020, to prevent an outbreak. Subsequently, with the increase in COVID-19 cases, Japanese medical students were also affected by the cancellation of clinical training and the shift to online education [10]. Due to the need to provide online lectures in almost all schools, the Japanese government established an exception to permit all educational institutions to use copyrighted works in online lectures free of charge only for the fiscal year 2020. Owing to this act, most educational institutions, including medical schools, shifted from traditional face-to-face education to online education, albeit with a few obstacles [5,11,12]. The number of people infected with COVID-19 was high not only in Japan; European countries also had high levels of infection [13]. Accordingly, the nationwide lockdown restrictions were implemented to control the spread of disease in Europe. Medical education in Europe has also been affected by the halting of lectures and clinical training, resulting in a sudden shift to online teaching [14,15,16].

Given that several studies have shown that the novel learning methods are equally or more effective than the traditional offline learning in ordinary settings [17,18,19,20,21], in an emergency setting such as the COVID-19 pandemic, online education potentially could be helpful and more effective to continue education in medical schools. However, the first-hand experience of medical students with regard to online education under such emergency conditions has yet to be evaluated. Accordingly, this study chiefly aimed to elucidate and discuss the impact of online learning among medical students in Japan during the COVID-19 pandemic. We included a few medical students from European countries with incomes as high as those from Japan because we intended to evaluate the results in Japan relatively [22].

## 2. Methods

### 2.1. Study Populations

Participants were recruited using snowball and purposive sampling methods as we wanted to recruit students who could afford to take part in this study, i.e., those whom we judged to be mentally well (not severely distressed by the shift to online learning). Data saturation was judged to be reached by all the authors: the collected data meaningfully responded to the research questions, and no further interviews would yield any additional insights to the study aim. We recruited participants from interviewers’ acquaintances and participants introduced the next one to us based on their acquaintances. We sent invitations to candidates using email and social networking services including LINE (LINE Corporation, Tokyo, Japan), Messenger (Facebook, Inc., Menlo Park, CA, USA), and Instagram (Facebook, Inc., Menlo Park, CA, USA). The study protocol was also made available to some participants who requested more detailed information. A total of 18 participants were included in the study. We stopped the interview (after obtaining a response from the 18th participant) once the data was found saturated since more interviews would not yield any new findings.

### 2.2. Interview Procedures and Content

Semi-structured interviews were conducted with medical students in Japan and overseas using web conferencing software ZOOM (version 5.4.6, Zoom Video Communications, Inc., Menlo Park, CA, USA). We used semi-structured interviews instead of structured ones. Using semi-structured questions as the basis, we allowed the topic to change, developed further, and summarized the items that the authors consider important and relevant since no similar study has been conducted in COVID-19 pandemic in the previous literature. The interviews were recorded. We employed a video interview method for several reasons such as international participant recruitment, adherence to physical distancing, and effectiveness for discussion [23,24].

There were six interviewers who are all Japanese medical students. All of them received two hours of training on how to effectively collect qualitative data from experts. Each interview was conducted by two interviewers to limit biases in questioning and interpretation. For each participant, two interviewers were chosen in such a way that the participant did not know anything about the interviewers, apart from that they were medical students. Before the interviews, we obtained the consent of all the participants to record their interviews for the analysis, and all interviews were recorded. 

The interview guide was developed based on our study interest and previous studies [5,17,19,20,25]. The interviewers were guided by the following five open-ended questions structured around the participants’ experience, mainly with online education during the COVID-19 pandemic: (Q1) a type of online education, which participants had received during the COVID-19 pandemic; (Q2) advantages and disadvantages of online education, which participants had experienced; (Q3) changes in the relationship with teachers, peers, friends, and family; (Q4) opinions on further improvement of online education, and (Q5) the necessity to belong to a particular university when education can be received online without belonging. Along with open-ended questions, participants were also asked for demographic information. To ensure the accuracy of transcribing, all participants received the transcript and confirmed the accuracy.

### 2.3. Interviewees

We sent the survey invitation and 21 participants agreed to be interviewed. In total, 18 participants finally completed an online interview via ZOOM: three did not complete due to system errors. Since data saturation was observed in the 18th interview, we did not feel the need to follow up with those three participants again. The interviews were conducted from 23 September to 3 October 2020.

### 2.4. Basic Characteristics

Participants were asked sex, nationality, country of university, age, grade, living status, academic confidence, a common place to study, study mode (whether they preferred studying alone, in a group, or both), previous experience of online learning, confidence in using the internet in general, and internet literacy (Table 1).

### 2.5. Data Analysis

Thematic analysis was conducted. Following the methods designed by Baraun and Clarke [26], after familiarization, codes were generated, which were then categorized thematically. Coherence and distinctiveness of themes were reviewed by A.O., Y.K., Y.S., and A.M.

### 2.6. Ethics Approval

The Institutional Review Board of Medical Governance Research Institute granted ethics approval of this study (MG2020-09-20200904) on 4 September 2020, adhering to the guidelines established by the Ministry of Health Labor and Welfare (MHLW) and The Ministry of Education, Culture, Sports, Science, and Technology (MEXT) in Japan [27].

## 3. Results

The mean interview time was 38 min (Min 27 min, Max 59 min, SD 18 min). The study participants were six males (33%) and 12 females (67%), and a median of 23 years old (IQR: 20–24). In total, 13 participants were from nine different medical schools in Japan, and the remaining five were from three in Slovakia, two in Norway and Hungary. Table 1 summarized the detailed characteristics of the participants and the interviews. 

## 4. Thematic Analysis

Participants’ responses to the five questions were coded and themed. 

### 4.1. Theme 1: Timesaving and Flexibility

Participants reported that online learning helped them to save a lot of time. This allowed them more time for other activities such as saving their commuting time or spending more time sleeping. In addition, some universities provided an environment in which students can watch online classes at any time by downloading recordings of real-time classes on their online platform. It gives medical students time flexibility in their study pace.

Participant No.13: “As I don’t have to commute to school, so I can sleep longer. I don’t have to go to and from school, so I can do whatever I want.” (Q2)

Participant No.10: “It takes a long time for students in rural areas to commute to school. … I can use that time to study for my exam.” (Q2)

Participant No.3: “I have lectures at the university hospital, so the classrooms are different every time. In online learning, I don’t have to spend time traveling and searching.” (Q2)

Participant No.2: “The advantage of recorded online classes is that they can be played back, rewound, and watched over and over again.” (Q2)

Participant No.8: “With online, we can study at our own pace. I don’t have to go to the university at a certain time.” (Q2)

### 4.2. Theme 2: Technical Problems and Lack of Digital Skills

Participants also reported the disadvantages of online learning such as technical problems. More than half of the participants reported these as a serious issue and that they had either encountered or witnessed some technical problems with accessing the course materials and tutoring sessions.

Participant No.6: “Some students even failed an exam because of technical errors; however, it wasn’t their fault. It was partly due to a lack of proper management by the university.”

Participant No.12: “There were a lot of problems. For example, the lecturer’s microphone wasn’t on, but he just kept talking without realizing it.” (Q1)

Participant No.16: “I had trouble getting into ZOOM.” (Q2)

Participant No.2: “The unstable Wi-Fi at my home put me in a tough situation. Because of it, my test was rescheduled.” (Q1)

Technical problems can be diverse, including system errors, the Wi-Fi setting at home, and the use of visual/audio devices. Some of these problems are related to the digital skills of the students and staff. Regardless of how good an online learning environment is, if it’s not used properly, students may encounter multiple problems.

### 4.3. Theme 3: Unstandardized Teaching Skills 

Many students observed great variation in the online teaching skills of their lecturers.

Participant No.8: “There were not so many well-designed classes. Many teachers were unfamiliar with online teaching, and only a few of them were making well-planned online classes.” (Q1)

Participant No.9: “The quality of the classes varied greatly from subject to subject, and I felt that regular face-to-face classes were more consistent and better.” (Q2)

Participant No.13: “Sometimes, online conversations were hard to understand. I prefer textbooks to facilitate learning.” (Q3)

While some lecturers were able to adjust to online teaching, others had trouble adjusting to a new method of teaching. During the COVID-19 pandemic, lecturers were forced to teach online even if they were unprepared for it, highlighting the lack of training they had in online teaching.

### 4.4. Theme 4: Lack of Experience beyond Medical Schools Lectures

Some students reported that they were not able to connect with faculty and friends due to restrictions imposed on them from going to the universities and hospitals as well as participating in extracurricular activities. Many participants commented that online classes made it difficult for both teachers and students to ask questions and to remember each other’s faces and names, which created a great distance from lecturers.

Participant No.3: ”Professors didn’t see us during the lecture because we turned off the camera in a big group.”

Participant No.11: “In online classes, faculty members no longer randomly call students to ask questions during class or have a small talk with students before or after class. As a result, the distance between the lecturers and students has increased, and I can’t match the face and name of some of the lecturers anymore.” 

Participant No.9: “As I couldn’t see my friends in the remote class, I wasn’t aware how they were preparing for the exam.” (Q2)

Participant No.3: “Before the summer I was quite lonely. It was hard to stay motivated to study since I wasn’t seeing my friends.” (Q3)

Students felt less connected with their peers, which led to a sense of loneliness and decreased motivation to study. Moreover, their inability to exchange information about the class and their progress also hindered them from staying motivated.

As for the benefits of belonging to a particular university (Question 5), many students mentioned the benefit of having a sense of belonging through relationships with friends and faculty through research and extracurricular activities.

Participant No.2: “The feeling of being in the part of group is one of the most important things. You go there every day and there are people to talk with friends.” (Q5)

Participant No.16: “If you want to be involved in cutting-edge research, I think it is worthwhile to belong to a particular university [that is strong in that research area], but I don’t think the university classes themselves have any value.” (Q5)

Although these students mentioned the advantages of belongingness to a particular university, these benefits such as participating in extracurricular activities and building relationship with friends and faculty were lessened in online education in the COVID-19 pandemic.

## 5. Discussion

This qualitative study evaluated online medical education in Japan and Europe from the students’ perspectives. Our thematic analysis identified four themes, illustrating the positive and negative aspects of online medical education during the COVID-19. 

One notable finding was that online learning saved time and provide flexibility to the students (Theme 1). The finding of our study is consistent with pre-COVID-19 online education studies. For example, a previous study in the United Kingdom reported that online learning saves students’ commuting time, allows students to study at their own pace, and gives wider access to learning materials as advantages [28]. Although this advantage has been noted in another previous study [29], our data suggests that this advantage was achieved even in the sudden enforced and unprepared delivery of online learning in the COVID-19 pandemic. 

While participants emphasized time-saving and time-flexibility as the biggest advantages, it is important to note that this may also imply an increased need for self-management. For example, students who usually maintained their academic motivation by meeting their teachers and friends in person (Theme 4) complained of decreased motivation to learn along with mental health difficulties. Previous studies in the U.S. and Japan have also reported an increase in the number of students suffering from psychological symptoms during the pandemic [25,30]. To support the learning of these students, the university’s staffs and wellbeing services need to be mindful of students’ mental health difficulties such as low motivation and loneliness. For example, mental health interventions such as social support to augment resilience and hope were reported helpful to online students [31]. These interventions need to be implemented and evaluated, in order to protect students from mental health difficulties.

In addition, a previous study has highlighted that living far from home was one of the risk factors for mental illness among students [30]. Considering such students who struggled with self-management and are at risk of mental illness, online education may require mental health support. During the COVID-19 pandemic, however, many students returned to their hometown or other socially supportive environments and studied from there, which might have mitigated the mental health difficulties associated with online learning. In line with the positive opinion of saving commuting time (Theme 1) and a negative association between commuting time to school/work and stress [28,32], we believe that time-flexibility such as saving commuting time and the previous report can be regarded as positive opinions that support the active introduction of online classes.

The negative themes include technical problems and a lack of digital skills (Theme 2), unstandardized teaching skills (Theme 3), loss of experiences other than lectures (Theme 4), which were also generally consistent with previous studies about online medical education before and during the COVID-19 pandemic [33,34]. The major difference between the pre-COVID-19 studies and the present study is that online medical education in the COVID-19 pandemic was introduced without sufficient preparation, leading to a great burden for many universities, faculty members, and students. This lack of preparation can somewhat contextualize these negative themes in this study. For example, some students reported that the quality of online education was dependent on the faculty members’ online teaching skills (Theme 3). Universities and faculty members, who had been active in online education during the pre-COVID-19 time have smoothly provided high-quality online education even in the COVID-19 pandemic [25]. Furthermore, sufficient training and preparation including the development of digital skills for online education can positively affect student satisfaction [35,36]. Along with those hard skills in online teaching, soft skills such as active interaction or an informal chat with students, in the online platform need to be developed [37], considering the comments such as “difficulty in asking questions”, “faculty members not remembering students’ faces”, and “no interaction during class” (Theme 4). These findings are in line with the social presence theory of online education, where the faculty’s presence in the curriculum can impact students’ learning experience [33,38].

In this respect, we believe that the social presence theory can be applied to medical education during the COVID-19 pandemic. It is difficult to bring about a sense of belongingness and community in online learning, which often leads to a feeling of alienation in students and lower learning outcomes. Indeed, the students who participated in this study accorded with it (Theme 4). Studies reported that emotional responses including politeness, humor, cooperative responses, and cohesiveness have been proposed as solutions, creating a stronger sense of community online [39]. We suggest that these attempts should be incorporated into online medical education during and possibly after the COVID-19 pandemic to increase student learning experience, achievement, and the meaning of belongingness in a particular university. 

Lastly, standardized teaching skills (Theme 3) can be another major solution to support students’ learning and wellbeing. For Theme 3: Unstandardized teaching skills, two possible solutions can be suggested based on our findings. The first is the need to create and distribute systematized materials, in which the order of teaching items is standardized (e.g., from diseases or from anatomy). Furthermore, these materials need to be prepared in the written format as well as the spoken one in online classes, which can address the auditory difficulties as reported by No. 13 in Theme 3: “Sometimes it is difficult to hear the teacher’s voice and words online.” This is often attributed to Theme 2: “Technical problems and digital skills.” Even when such problems occur, students can learn on their own if there are materials that describe the contents of the class. Second, faculty members should have the opportunity to contact students individually via email or other means to answer questions. According to our result in Theme 4, Participant No. 11 commented that students and faculty no longer talked to each other before or after class. Online classes are less engaged, especially with faculty members. As medical classes are relatively information-intensive, it is also important for online medical education to provide individualized communication before and after class. 

Finally, there was no significant difference in the results of the survey on online medical education between European medical students and Japanese medical students. We consider that there could be three plausible explanations. First, medical training is largely the same between Europe and Japan. Japan originally introduced the modern concept of medicine from European countries such as the Netherlands and Germany in the Edo and Meiji Era more than 100 years ago. Second, the developmental level of the two regions is roughly the same as shown in the gross domestic product per capita of two regions [40]. Lastly, we could only include four participants from Europe, which might have limited the ability to detect potential differences between the two education systems. Indeed, this is a substantial limitation of this study and is later covered in the limitation section as well.

## 6. Limitations

First, the sample size for students from Europe was low. A larger and more balanced sample would have made our findings more generalizable. Second, the interview questions have the same structure and guidelines, but the duration of the interview was not constant. This might suggest variance in the depth of the participant responses.

## 7. Conclusions

The novelty of our study lies in positive aspects such as timesaving and flexibility in the learning environment, and negative ones such as internet condition, quality of online lectures and lack of experience beyond medical school lectures (e.g., extracurricular activities). Findings from this study will provide foundational data to improve the quality of online medical education.

## Figures and Tables

**Table 1 ijerph-19-02840-t001:** Basic characteristic.

Sex/No	Native Place	Country of University	Age	Grade	Interview Duration (min)	Living Status	Academic Confidence	Place to Study	Study Mode	Experience of Online Learning	Internet Confidence	Internet Literacy
F1	PNS	Slovakia	30	6	46	Alone	Good	Library	Alone	Yes	Poor	Yes
F2	Germany	Slovakia	26	6	34	Alone	Good	Library	Group	Yes	Poor	Yes
M3	Norway	Norway	25	6	39	Friends	Good	Library	Group	No	Poor	Yes
M4	Japan	Japan	24	6	20	Alone	Poor	Home, University	Both	No	Poor	Yes
M5	Japan	Japan	23	6	45	Alone	Good	Home	Alone	Yes	Good	Yes
F6	China	Hungary	25	4	30	Alone	Moderate	Library Home	Group	Yes	Poor	Yes
M7	Japan	Japan	23	4	30	Alone	Good	Home	Alone	No	Poor	Yes
F8	Japan	Slovakia	PNS	4	51	Alone	Poor	Home	Group	No	Poor	Yes
F9	Japan	Japan	23	4	31	Alone	Good	Home	Both	No	Poor	Yes
F10	Japan	Japan	23	4	45	Alone	Good	Home	Alone	Yes	Good	Yes
F11	Japan	Japan	22	3	27	Alone	Good	Library	Both	No	Poor	Yes
M12	Japan	Japan	21	3	30	Alone	Good	Library	Alone	No	Poor	No
M13	Japan	Japan	21	3	47	Alone	Good	Library	Group	No	Poor	Yes
F14	Japan	Japan	20	3	33	Alone	Good	Home	Alone	No	Poor	Yes
M15	Japan	Japan	20	3	30	Alone	Poor	Café	Alone	No	Poor	Yes
F16	Japan	Japan	20	2	35	Alone	Good	Library	Both	Yes	Poor	Yes
F17	Japan	Japan	20	2	59	Alone	Good	Home	Alone	Yes	Good	Yes
F18	Japan	Japan	19	1	46	Family	Good	Outside	Alone	Yes	Poor	Yes

PNS = Prefer Not to Say.

## Data Availability

The datasets not available due to ethical restrictions.

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
