# Peer review of "Cross-Country Student Perceptions about Online Medical Education during the COVID-19 Pandemic"

_ijerph, 2022, doi:10.3390/ijerph19052840_

Round 1

Reviewer 1 Report

The study was developed during the COVID-19 pandemic, including online semi-structured interviews with18 medical students in several countries. It aimed to identify perceptions for online medical education  with reference to the positive and negative aspects. It is, inevitably, a current topic with potential to reflect on the quality of experiences in online environments for this particular group of students. 

Nevertheless, authors do not suggest how to improve the quality of online medical education. In terms of pedagogical approach, it would be interesting a deeper understanding of what authors referred as standardized teaching skills (Theme 3) since it can be seen as a major solution to support students’ learning and wellbeing. Some other aspects related to the diversity of participants' countries of origin could be explored, focusing particularly the relationship between these specificities and the perception of students in terms of difficulties and potential within online environments.

Author Response

Reviewer 1

The study was developed during the COVID-19 pandemic, including online semi-structured interviews with18 medical students in several countries. It aimed to identify perceptions for online medical education with reference to the positive and negative aspects. It is, inevitably, a current topic with potential to reflect on the quality of experiences in online environments for this particular group of students. Nevertheless, authors do not suggest how to improve the quality of online medical education. In terms of pedagogical approach, it would be interesting a deeper understanding of what authors referred as standardized teaching skills (Theme 3) since it can be seen as a major solution to support students’ learning and wellbeing.

Thank you for your suggestion. I have added the following to the discussion.

(Page 8, line 355 to line 370 in the revised manuscript)

For Theme 3: Unstandardized teaching skills, two possible solutions can be suggested based on our findings. The first is the need to create and distribute systematized materials, in which the order of teaching items is standardized (e.g., from diseases or from anatomy). Furthermore, these materials need to be prepared in the written format as well as the spoken one in online classes, which can address the auditory difficulties as reported by No. 13 in Theme 3: "Sometimes it is difficult to hear the teacher's voice and words online”. This is often attributed to Theme 2, "Technical problems and digital skills". Even when such problems occur, students can learn on their own if there are materials that describe the contents of the class.

Second, faculty members should have the opportunity to contact students individually via email or other means to answer questions. According to our result in Theme 4, Participant No. 11 commented that students and faculty no longer talked to each other before or after class. Online classes are less engaged, especially with faculty members. Because medical classes are relatively information-intensive, it is also important for online medical education to provide individualized communication before and after class.

Some other aspects related to the diversity of participants' countries of origin could be explored, focusing particularly the relationship between these specificities and the perception of students in terms of difficulties and potential within online environments.

Thank you for your important suggestion. We have now added the following sentences to the discussion.

(Page 8, line 371 to line 399in the revised manuscript)

Finally, there was no significant difference in the results of the survey on online medical education between European medical students and Japanese medical students. We consider that there could be four plausible explanations for the lack of difference. First, key trainings of medicine are largely the same. Second, key characteristics of medical education may be consistent between Europe and Japan. Japan originally introduced the modern concept of medicine from European countries such as the Netherlands and Germany in Edo and Meiji Era more than 100 years ago. Thus, we speculate that the online education of medicine in each region may have become a similar product. Thirdly, the developmental level of two regions are roughly the same as shown in the gross domestic product per capita of two regions[40].  Lastly, we could only include two participants in Europe. Consequently, we could not have detected potential differences between the two education systems. This is a substantial limitation of this study and is later covered in the limitation section as well.

Reviewer 2 Report

The research topic is timely and the research results are considered to have historical value.

Abstract
- It seems that much of the content of the conclusion should be moved to the result section. It is better to write the conclusion very briefly than it is now.

Introduction
- In addition to Japanese medical students, the study subjects also include medical students from Slovakia, Norway, and Hungary. But, most of the introduction is described mainly about the COVID-19 situation in Japan. Interview-style research reflects their experiences a lot. Therefore, it is necessary to understand their situation. Therefore, in the introduction, like Japan, the situation in Europe should be more descriptive. 

Methods
- Given that the small sample size is a limitation of the study, you would have expected that the sample size would be a problem, but why did you use the snowball sampling method? If authors plan a joint research project with your university and medical schools in other countries, it would be possible to conduct a survey including more medical students.

- Authors said the question was structured. What was the rationale for making this? For example, was it based on literature, or was it selected through expert opinion or advice?

Result
- Although the interview questions have the same structure and guidelines, the duration of the interview is not constant. This can also affect bias.

Review
- (Especially) Theme 1 and Theme 4 would have shown this result regardless of the rest of the themes, even if they were not medical students but students of other majors. However, Theme 2 and Theme 3 may differ a lot depending on how well you use the Internet et al.. This is the same for students as well as professors. It also depends on how well the university provides technical support for these areas. Therefore, it is thought to be cautious in generalizing as this is considered to be highly dependent on the characteristics of the study subjects.

Author Response

Dear

Reviewer 2

Abstract
- It seems that much of the content of the conclusion should be moved to the result section. It is better to write the conclusion very briefly than it is now.

Thank you for pointing this out. In the latest version, we have revised the manuscript as follows.

(Page 9 line 401 to line 405 in the revised manuscript)

  1. Conclusions

Novelty of our study lies in positive aspects such as timesaving and flexibility in the learning environment and negative ones such as internet condition, quality of online lectures and lack of university experiences outside of the lectures (e.g., extracurricular activities). Findings from this study will provide foundational data to improve the quality of online medical education.

Introduction
- In addition to Japanese medical students, the study subjects also include medical students from Slovakia, Norway, and Hungary. But, most of the introduction is described mainly about the COVID-19 situation in Japan. Interview-style research reflects their experiences a lot. Therefore, it is necessary to understand their situation. Therefore, in the introduction, like Japan, the situation in Europe should be more descriptive.

Thanks for pointing this out. In the latest version, we have also added European information to the introduction as follows.

(Page 2 line 58 to line 64 in the revised manuscript)

Not only in Japan but also in European countries, the number of people infected with COVID-19 was high []. Accordingly, the nationwide lockdown restrictions were implemented to control the spread of disease in many countries in Europe.Inevitably, medical education in Europe has also been affected, with the halting of lectures and clinical placements, resulting in a sudden shift to online teaching [14-16] . In order to evaluate the results in Japan relatively, we also included students from a few medical students in European countries with incomes as high as Japan’s.[17]

Methods
- Given that the small sample size is a limitation of the study, you would have expected that the sample size would be a problem, but why did you use the snowball sampling method? If authors plan a joint research project with your university and medical schools in other countries, it would be possible to conduct a survey including more medical students.

  (Page 2 line 75 to line 79 in the revised manuscript)

Thanks for pointing this out. I would like to explain two reasons why we used snowball sampling. First, our research was a qualitative study and data saturation was reached. Second, we wanted to recruit students who could afford to take part in this study, i.e., those whom we judged to be mentally well (not severely distressed by the shift to online learning).

- Authors said the question was structured. What was the rationale for making this? For example, was it based on literature, or was it selected through expert opinion or advice?

(Page 3 line 117 to page XX line 121 in the revised manuscript)
We used semi-structured interviews instead of structured ones. Using semi-structured questions as the basis, we allowed the topic to change, developed further and summarized the items that the authors consider important and relevant, since no similar study has been conducted in COVID-19 pandemic in the past literature.

Result
- Although the interview questions have the same structure and guidelines, the duration of the interview is not constant. This can also affect bias.

Thank you for pointing this out. This has now added to the Limitations.

(Page 9 line 393 to line 399 in the revised manuscript)

  1. Limitations

First, since our sample size was modest, combining Japanese and fewer European students, a larger and more balanced sample can address generalizability of our findings. Second, since this study was conducted only in medical school and no comparison was made with students in other subjects, themes that are unique to medical students were not identified. Third, the interview questions have the same structure and guidelines, however the duration of the interview was not constant. This might suggest variance in the depth of the participant responses.

Reviewer 3 Report

This article presents questionable results in terms of quality as such a low number of respondents (18 students) raise serious concerns about the study’s statistical significance. In addition, the article lacks any quantitative analysis which could benefit to article’s scientific significance however in that case it would require validation and approval of the study's protocol from the ethical committee side.

N.B. It is not acceptable to publish a table in the form it is in the article.

Author Response

Dear Reviewer 3

This article presents questionable results in terms of quality as such a low number of respondents (18 students) raise serious concerns about the study’s statistical significance. In addition, the article lacks any quantitative analysis which could benefit to article’s scientific significance however in that case it would require validation and approval of the study's protocol from the ethical committee side. 

N.B. It is not acceptable to publish a table in the form it is in the article.

Thank you for your helpful comment. We would agree that our sample size of 18 is not large enough to consider inferential statistics, however in this manuscript we only reported descriptive statistics. As you noted, this was not clear in the previous version, therefore it is now clarified (p.2, L.76). Moreover, the study including the methodology was approved by the Institutional Review Board of Medical Governance Research Institute granted ethics approval of this study (MG2020-09-20200904), and it is now more clearly reported in the manuscript. Table is now re-formatted, thank you.

Round 2

Reviewer 2 Report

It seems that the previous deficiencies have been corrected.

Author Response

Thank you for your comment.

Reviewer 3 Report

I appreciate the author’s work, but corrections and clarifications should be introduced into the manuscript, as well as extensive editing of the English language and style required prior to publication.

Line 32: There is no need to capitalize the letter in “Experiences“.

Line 41: Remove the extra space after “(COVID-19)”.

Line 66: The space is needed after “[18–22]”.

Line 97: The comma is required after “physical distancing”.

Line 106: The space is needed after “[5,18,20,21,25]”.

Line 106: A definitive article is needed before “following”.

Line 160: Remove the comma after “as”.

Line 177: The sentence has to be rewritten to make sense.

Line 178: According to English grammar, the word “particularly” must be replaced with “particular”.

Line 182: Remove the comma after “issue”.

Line 226: The sentence has to be rewritten to make sense.

Lines 242 - 246: The section has to be rewritten to make sense.

Line 248: According to English grammar, the part “of a” must be replaced by “to a”.

Line 249: The “in” is needed after “participating”.

Line 261: According to English grammar, the word “other” must be replaced by “another”.

Line 293: The comma is required after “members”.

Author Response

Reviewer 3

I appreciate the author’s work, but corrections and clarifications should be introduced into the manuscript, as well as extensive editing of the English language and style required prior to publication.

In line with your comment, our corrections and clarifications are embedded in the manuscript. English language and formatting have been carefully reviewed.

Line 32: There is no need to capitalize the letter in “Experiences“.

(Page 1, Line 32)

In line with your comment, now the lower-case is used.

Line 41: Remove the extra space after “(COVID-19)”.

(Page2, Line 50)

We have removed the extra space after “(COVID-19)”.

Line 66: The space is needed after “[18–22]”.

(Page 2, Line 75)

Now a space is added.

Line 97: The comma is required after “physical distancing”.

(Page3 Line 148)

We have added a comma after “physical distancing”.

Line 106: The space is needed after “[5,18,20,21,25]”.

(Page 3, Line157)

We have added a space after” “[5,18,20,21,25]”.

Line 106: A definitive article is needed before “following”.

(Page 3, Line157)

We have added “the” before “following”.

Line 160: Remove the comma after “as”.

(Page 5, Line 262)

We have removed the comma after “as”.

Line 177: The sentence has to be rewritten to make sense.

(Page5 Line 215)

This sentence has been removed because it duplicated the information in lines 261-266.

Line 178: According to English grammar, the word “particularly” must be replaced with “particular”.

This sentence with the word ”particularly” was removed because it duplicated the information in lines 338-342.

Line 182: Remove the comma after “issue”.

(Page5, Line 280)

We have removed the comma after “issue”.

Line 226: The sentence has to be rewritten to make sense.

This sentence in page 6 line 261-262 in the previous version has been removed because it duplicated the information in lines 338-342 in the newest version.

Lines 242 - 246: The section has to be rewritten to make sense.

We have rewritten the section as follows.

(Page 7, Line 364-366)

Participant No16: “If you want to be involved in cutting-edge research, I think it is worthwhile to belong to a particular university [that is strong in that research area], but I don't think the university classes themselves have any value.” (Q5)

Line 248: According to English grammar, the part “of a” must be replaced by “to a”.

(Paeg7 Line 495)

We have replaced “of a” with “to a”.

Line 249: The “in” is needed after “participating”.

(Page7, Line 496)

We have added “in” after “participating”.

Line 261: According to English grammar, the word “other” must be replaced by “another”.

(Page 7 Line 509)

We have replaced “other” with “another”.

Line 293: The comma is required after “members”.

(Page 7 Line 543)

We have added a comma after “members”.